# Preparation of Two-Dimensional Layered CeO₂/Bi₂O₃ Composites for Efficient Photocatalytic Desulfurization

**Xiaowang Lu** [1,2,*], **Wenxuan Chen** [1], **Haijun Hou** [1,2], **Junchao Qian** [3] and **Qinfang Zhang** [1,*]

1   School of Material Science and Engineering, Yancheng Institute of Technology, Yancheng 224051, China
2   Jiangsu Provincial Key Laboratory of Eco-Environmental Materials, Yancheng Institute of Technology, Yancheng 224051, China
3   School of Material Science and Engineering, Suzhou University of Science and Technology, Suzhou 215009, China
*   Correspondence: luxiaowang@ycit.edu.cn (X.L.); qfangzhang@gmail.com (Q.Z.)

**Abstract:** A two-dimensional layered $CeO_2/Bi_2O_3$ composite was synthesized by microwave solvothermal method. X-ray diffraction (XRD), Raman spectroscopy, field emission scanning electron microscope (FESEM), transmission electron microscope (TEM), UV-Vis diffuse reflection spectroscopy (DRS), and X-ray photoelectron spectroscopy (XPS) were used to studied crystal structure, morphology, optical performance, elemental composition and the surface electronic state of the samples. The photocatalytic properties of the prepared samples were evaluated by photocatalytic desulfurization under visible light. When the molar ratio of Ce and Bi was 1:2, $CeO_2/Bi_2O_3$ composite presented the highest photocatalytic desulfurization rate. Transient Photocurrent measurement, electrochemical impedance spectroscopy (EIS) and photoluminescence spectroscopy (PL) showed that $CeO_2$ and $Bi_2O_3$ formed a heterojunction, which could promote the separation of photogenerated electrons and holes, improving the photocatalytic activity. Furthermore, it was found that the active species of hydroxyl radical (·OH) played an important role in the photocatalytic degradation of dibenzothiophene (DBT) based on the active species capture experiment. Finally, a plausible mechanism for the photocatalytic oxidative desulfurization of this nanocomposite was proposed.

**Keywords:** $CeO_2$; $Bi_2O_3$; composite; heterojunction; photocatalytic desulfurization





## 1. Introduction

Nowadays, environmental pollution and climate change caused by fossil energy combustion have become the focus of global attention. In particular, the combustion of sulfides in motor vehicle fuel generates a large amount of sulfur oxides. These sulfur oxides can cause acid fog, acid rain and haze, lead to environmental pollution and threaten human health [1,2]. Therefore, it is necessary to develop deep desulfurization technology to remove the sulfide in fuel oil and decrease the emission of sulfur oxides from the source. At present, hydrodesulfurization (HDS) is the main technology, widely used in modern industry to reduce the sulfur content of fuel. It requires high pressure, high temperature and high hydrogen consumption. Dibenzothiophene (DBT) and its derivatives are the main sulfur species in diesel and gasoline. However, HDS technology experiences difficult in removing them [3,4]. In order to solve these problems, several hydrogen-free desulfurization technologies have been developed, such as oxidative desulfurization (ODS) [5], adsorptive desulfurization (ADS) [6], extractive desulfurization (EDS) [7], chemical desulfurization (CDS) [8] and biological desulfurization (BDS) [9], etc. Among these, the ODS method [10], as a deep desulfurization technology, has attracted wide attention because it can selectively convert organosulfur compound into corresponding sulfones and sulfate ions under moderate reaction conditions. In recent years, the photocatalytic oxidation desulfurization (PODS) method has shown extraordinary potential due to its special advantages, such as carbon neutrality, chemical reactions driven by photons instead of high pressure and high temperature,

and the utilization of abundant available solar energy [11,12]. In this technique, oxidizing reagents such as molecular oxygen($O_2$) [13], hydrogen peroxide($H_2O_2$) [14], ozone ($O_3$) [15], and air [16] are used to convert organosulfur compounds present in motor fuels into their corresponding sulfones or $SO_4^{2-}$, which can be removed through adsorption or extraction to achieve deep desulfurization. In addition, the activity of photocatalytic reactions mainly depends on the photocatalyst, so photocatalytic materials need to have stable structures and excellent optical properties Therefore, a series of excellent photocatalysts for photo-catalytic desulfurization have been developed. Hussain's group developed a deep aerobic photocatalytic oxidation desulfurization technology and synthesized a series of excellent photocatalysts such as $LaVO_4$ [17,18], $Ag_3VO_4$ [19], $V_2O_5$ [20], etc.; Zhou et al. [21] prepared a $Ag_2O/Na$-g-$C_3N_4$ heterojunction for photocatalytic desulfurization of thiophene in fuel. Belousov et al. [22] synthetized nanosized $Bi_2W_xMO_{1-x}O_6$ solid solutions to degrade DBT in model fuel; Zhang et al. [23] prepared a $CeF_3/g$-$C_3N_4$ heterojunction photocatalyst with up-conversion performance for photocatalytic removal of DBT from model oil.

Currently, a rare earth oxide ($CeO_2$) is considered as one of the semiconductor materials with various application prospects, especially in the field of photocatalysis, due to its unique 4f orbital and abundant electronic energy levels. However, single $CeO_2$ still has some disadvantages, such as large band gap, low absorption in visible region and high electron–hole recombination. Nevertheless, these shortcomings can be overcome by coupling other semiconductors, promoting charge separation and increasing carrier lifetime to improve photocatalytic activity [24]. Therefore, a series of Ceria-based photocatalytic composites have been developed for photocatalytic desulfurization. Mousavi-Kamazani et al. [25] synthetized $Cu_2O$-$CeO_2$ nanocomposites, which has 84% photocatalytic desulfurization efficiency within 180 min under visible light; Radwan et al. [26] prepared $Fe_2O_3$-$CeO_2$ nanocomposites, and research showed that, when the loading amount of $Fe_2O_3$ was 15%, the composite had the highest photocatalytic desulfurization activity; Chen's group [27] prepared well-aligned a $CeO_2/TiO_2$ nanotube array photocatalyst and, when used for photocatalytic oxidation of benzo-thiophene (BT) under visible light irradiation, it was found that more than 90% of sulfur compounds in model oil were removed.

The semiconductor oxide $Bi_2O_3$ has recently attracted the attention of researchers due to its unique optical and electrical properties. It is well known that bismuth oxide has six crystal forms, denoted as $\alpha$-$Bi_2O_3$ (monoclinic), $\beta$-$Bi_2O_3$ (tetragonal), $\delta$-$Bi_2O_3$ (cubic bcc), $\gamma$-$Bi_2O_3$ (orthorhombic), $\varepsilon$-$Bi_2O_3$ (orthorhombic), and $\omega$-$Bi_2O_3$ (triclinic), respectively. Among them, $\alpha$-$Bi_2O_3$ is considered as a promising photocatalyst for water decomposition and organic photocatalytic degradation under visible light illumination, with good stability, non-toxicity and small band gap [28]. Dong et al. [29] successfully prepared a porous nanosheet structure $\alpha$-$Bi_2O_3$ photocatalyst by the biomimetic-synthesis assisted hydrothermal method, which displayed efficient degradation toward different pollutant molecules; Gupta et.al. [30] synthesized $\alpha$-$Bi_2O_3$ nanosheets by a simple annealing assisted thermal decomposition method, which exhibited high photocatalytic degradation of Rhodamine B. Therefore, $CeO_2$ coupled with $\alpha$-$Bi_2O_3$ to construct a heterojunction structure is expected to improve photocatalytic activity. In addition, two-dimensional (2D) semiconductor photocatalysts have been widely investigated due to their good light absorption characteristics, shorter electron and hole migration paths [31,32]. Lu et al. [33] synthesized N-doped two-dimensional $CeO_2$-$TiO_2$ nanosheets by a biological template method, which exhibited excellent photocatalytic desulfurization activity; Li et al. [34] utilized attapulgite-$CeO_2$ decorated two-dimensional $MoS_2$ to prepare nanocomposite, and the degradation rate of DBT can reach 95% under 3 h irradiation.

Herein, in this paper, we prepared a two-dimensional layered $CeO_2/Bi_2O_3$ composite using a microwave solvothermal method. Then the activity of the photocatalysts were evaluated by removing DBT from the model oil. In addition, the crystal structure, morphology, optical performance, elemental composition and surface electronic state of the prepared samples were investigated by XRD, Raman spectroscopy, FESEM, TEM, DRS and XPS. Meanwhile, the separation efficiency of photogenerated electrons and holes was studied by

photocurrent measurement, EIS and PL. Finally, the possible mechanism of photocatalytic desulfurization was discussed.

## 2. Results and Discussion

### 2.1. XRD Analysis

Figure 1 shows the XRD patterns of the prepared photocatalysts. The diffraction peaks near 28.6°, 33.1°, 47.5°, 56.3° and 59.1° correspond to (111), (200), (220), (311) and (222) crystal planes of cubic fluorite structure $CeO_2$ (PDF#34-3094), respectively. Accordingly, the main characteristic peaks at 24.6° 25.8°, 26.9°, 27.4 °, 28.0 °, 33.0°, 33.2°, 34.0°, 35.0°, 37.6°, 40.0°, 42.4°, 45.1°, 46.3°, 48.6°, 52.4°,54.7°, 57.9°, 59.1°, 62.5° and 71.4 are indexed to (−102), (002), (−112), (−121), (012), (−122), (−202), (022), (−212), (−113), (−222), (−123), (023), (041), (−104), (−332), (−241), (024), (150), (104) and (−161) the crystal planes of monoclinic α-$Bi_2O_3$ (PDF#71-2274), respectively. In CeBi-2 and CeBi-1 composites, the main characteristic peaks are similar to $CeO_2$, and some low-intensity $Bi_2O_3$ characteristic peaks appear. However, in CeBi-0.5 composite, the main characteristic peaks of the composite are similar to α-$Bi_2O_3$, and some low-intensity $CeO_2$ characteristic peaks appear. It is worth noting that, with the increase of the $Bi_2O_3$ component, the characteristic diffraction peaks of the monoclinic α-$Bi_2O_3$ were gradually enhanced. Moreover, compared with pure $CeO_2$ and $Bi_2O_3$, some peak positions of the composites are slightly shifted, which is due to the strong interfacial interaction between $CeO_2$ and $Bi_2O_3$ [35]. This indicates that a heterostructure may be formed at the interface between $CeO_2$ and $Bi_2O_3$.

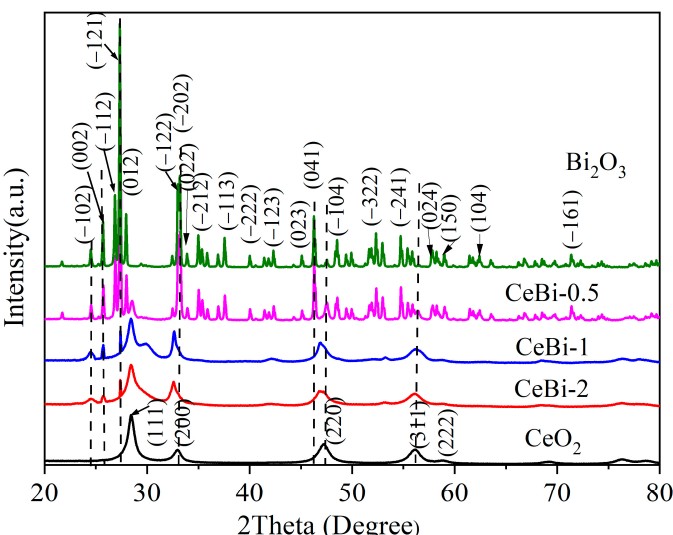

**Figure 1.** XRD patterns of $CeO_2$, $Bi_2O_3$ and $CeO_2/Bi_2O_3$ composites.

### 2.2. Raman Spectra Analysis

Figure 2 shows the Raman characteristic peaks of $CeO_2$, $Bi_2O_3$ and $CeO_2/Bi_2O_3$ composites in the wavenumber range of 100–650 cm$^{-1}$. The peak at 460 cm$^{-1}$ in the pure $CeO_2$ corresponds to the symmetrical stretching vibration of Ce-O-Ce, which belongs to the triple degenerate $F_{2g}$ vibration mode of $CeO_2$ in the cubic fluorite phase [36]. However, for pure α-$Bi_2O_3$, the characteristic peak centered at 117 cm$^{-1}$ is attributed to the vibration modes of bismuth atoms; while the characteristic peaks at 138 cm$^{-1}$ and 150 cm$^{-1}$ correspond to Bi-O stretching vibration modes; the bands at 183 cm$^{-1}$, 210 cm$^{-1}$, 284 cm$^{-1}$, 313 cm$^{-1}$, 410 cm$^{-1}$, 445 cm$^{-1}$ and 530 cm$^{-1}$ belong to the oxygen vibration modes [37]. Additionally, in the Raman spectra of $CeO_2/Bi_2O_3$ composites with different ratios, it can be found that the broad peaks of $CeO_2$ (445 cm$^{-1}$) and $Bi_2O_3$ (460 cm$^{-1}$) overlap. Meanwhile, it is observed that the characteristic peak intensity of $Bi_2O_3$ diminishes with the increase in $CeO_2$ content, all of which indicate that $CeO_2/Bi_2O_3$ composites are successful prepared.

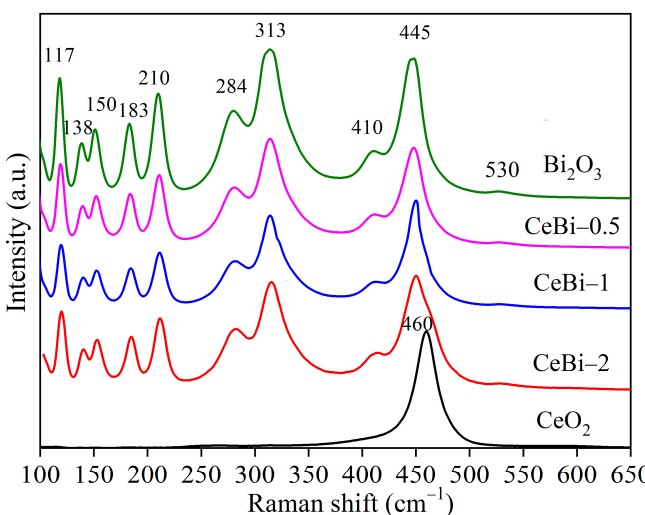

**Figure 2.** Raman spectra of $CeO_2$, $Bi_2O_3$ and $CeO_2/Bi_2O_3$ composites.

### 2.3. SEM, TEM, HRTEM and EDS Analysis

The morphology and microstructure of the prepared photocatalyst are studied by FESEM, TEM, HRTEM and EDS. Figure 3a,b show the FESEM images of CeBi-0.5. It can be seen that CeBi-0.5 presents an oval nanosheet structure with uniform shape and size on a large scale. The sizes of the longitudinal axis and horizontal axis are about 500 nm and 1 μm, respectively, and the nanosheets self-assemble into flower-like nanostructures. Figure 3c,d are the TEM and HRTEM images of CeBi-0.5. It is observed that the prepared composite presents a two-dimensional lamellar structure, which is composed of about 5–20 nm nanoparticles. In addition, the lattice fringes with an interplanar spacing of 0.32 nm and 0.27 nm correspond to (111) crystal planes of cubic fluorite $CeO_2$ and (200) crystal plane of monoclinic $\alpha$-$Bi_2O_3$, respectively (Figure 3d). Figure 3e shows the EDS spectrum of CeBi-0.5 composite. Only Bi, Ce and O elements are detected, and no other elements, indicating that the $CeO_2$ and $Bi_2O_3$ are successfully combined.

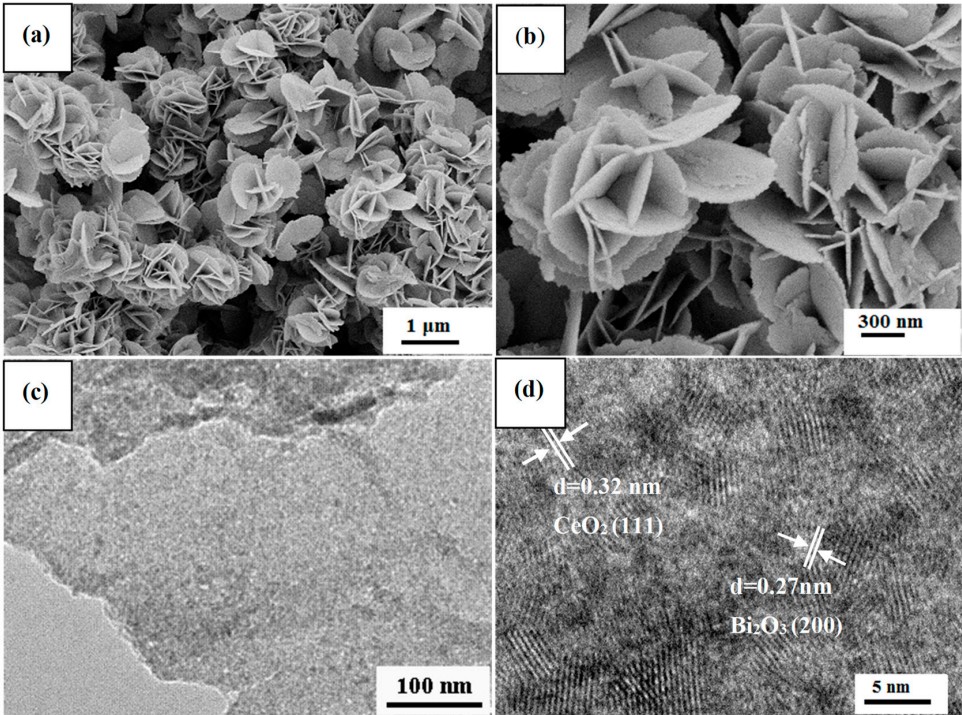

**Figure 3.** *Cont.*

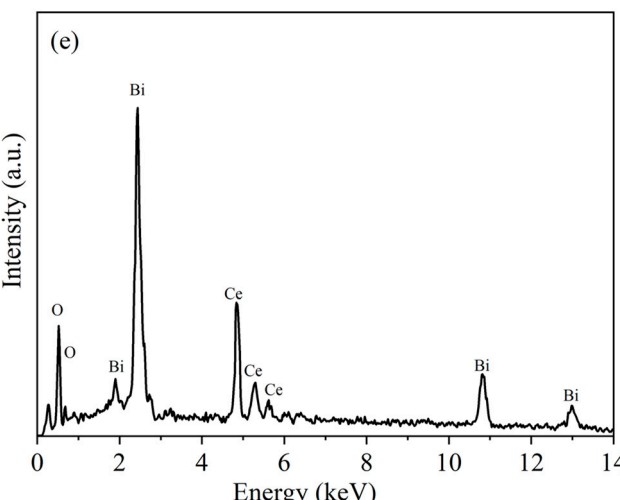

**Figure 3.** FESEM of images of CeBi-0.5 (**a,b**); TEM (**c**) and HRTEM (**d**) images of CeBi-0.5; EDS spectrum of CeBi-0.5 (**e**).

### 2.4. Nitrogen Adsorption–Desorption Isotherm and Pore Size Distribution

The specific surface area and pore-size distribution of the photocatalysts have a major influence on the enhancement of the photocatalytic performance. Figure 4 shows the $N_2$ adsorption–desorption isotherm of $CeO_2$, $Bi_2O_3$ CeBi-0.5 and the pore-size distribution isotherm of CeBi-0.5. The curve of pure $CeO_2$ presents an inverse S shape and there is no hysteresis loop, which belongs to a type II adsorption isotherm, indicating that pure $CeO_2$ has no pores. The $Bi_2O_3$ and CeBi-0.5 composite exhibit a typical IV like isotherm with H3 hysteresis loop, indicative of the characteristic of mesopores, according to the IUPAC classification [38]. It can be seen that the CeBi-0.5 composite has a relatively narrow pore size distribution, centered at about 5–20 nm (the inset of Figure 4). The BET-specific surface areas of $CeO_2$, $Bi_2O_3$ and CeBi-0.5 composite are measured as 4.8 $m^2/g$, 38.7 $m^2/g$ and 52.5 $m^2/g$, respectively. In particular, two-dimensional CeBi-0.5 composite exhibit higher surface areas than pure $CeO_2$ and $Bi_2O_3$, which may offer a more active site in the process of photocatalytic desulfurization, enhancing the photocatalytic performance.

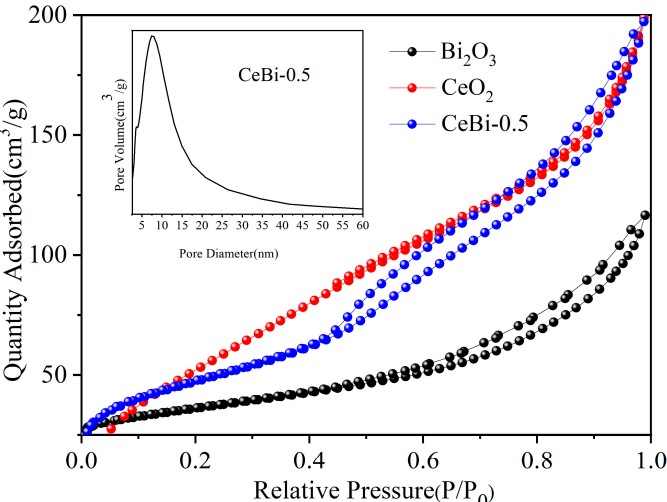

**Figure 4.** $N_2$ adsorption–desorption isotherm of $CeO_2$, $Bi_2O_3$ CeBi-0.5 and the pore-size distribution isotherm of CeBi-0.5.

## 2.5. XPS Analysis

The chemical states and bonding environment of the elements on the surface of the composites are analyzed by XPS. Figure 5a shows the high-resolution spectrum of the Ce 3d. The emission peaks of Ce 3d are divided into eight fitting peaks, where the v and u represent spin-orbit coupling $3d_{5/2}$ and $3d_{3/2}$, respectively. The peaks of $v_1$ and $u_1$ are assigned to the Ce IV ($3d^9 4f^2$) O ($2p^4$) state; the peaks of $v_3$ and $u_3$ are assigned to the Ce IV ($3d^9 4f^1$) O ($2p^5$) state and the peaks of $v_4$ and $u_4$ are assigned to the Ce IV ($3d^9 4f^0$) O ($2p^6$) state. These six fitting peaks belong to the $Ce^{4+}$ species. The other two fitting peaks $v_2$ and $u_2$ are related to the Ce III ($3d^9 4f^2$) O ($2p^5$) state, which belong to the $Ce^{4+}$ species. Therefore, there are trivalent and tetravalent cerium in $CeO_2$ and CeBi-0.5 [39]. The presence of trivalent cerium implies the presence of more oxygen vacancies. As shown in Figure 5b, there are characteristic peaks at 159 eV and 164.3 eV, which are attributed to Bi 4 $f_{7/2}$ and Bi 4 $f_{3/2}$ spin orbitals, respectively [40]. This indicates the presence of trivalent bismuth in $Bi_2O_3$ and CeBi-0.5. The fitting peaks located at 529.7 eV (529.5 eV) and 531.3 eV (531.2 eV) observed in Figure 5c correspond to the lattice oxygen and surface hydroxyl oxygen in $CeO_2$ ($Bi_2O_3$), respectively. It is noted that, compared with pure $CeO_2$ and $Bi_2O_3$, the binding energies of Ce 3d, Bi 4f and O 1s have slightly shifted, which indicates that there is a strong electron interaction between the $CeO_2$ and $Bi_2O_3$ interfaces [41]. Therefore, the heterojunction structure between $CeO_2$ and $Bi_2O_3$ may be formed.

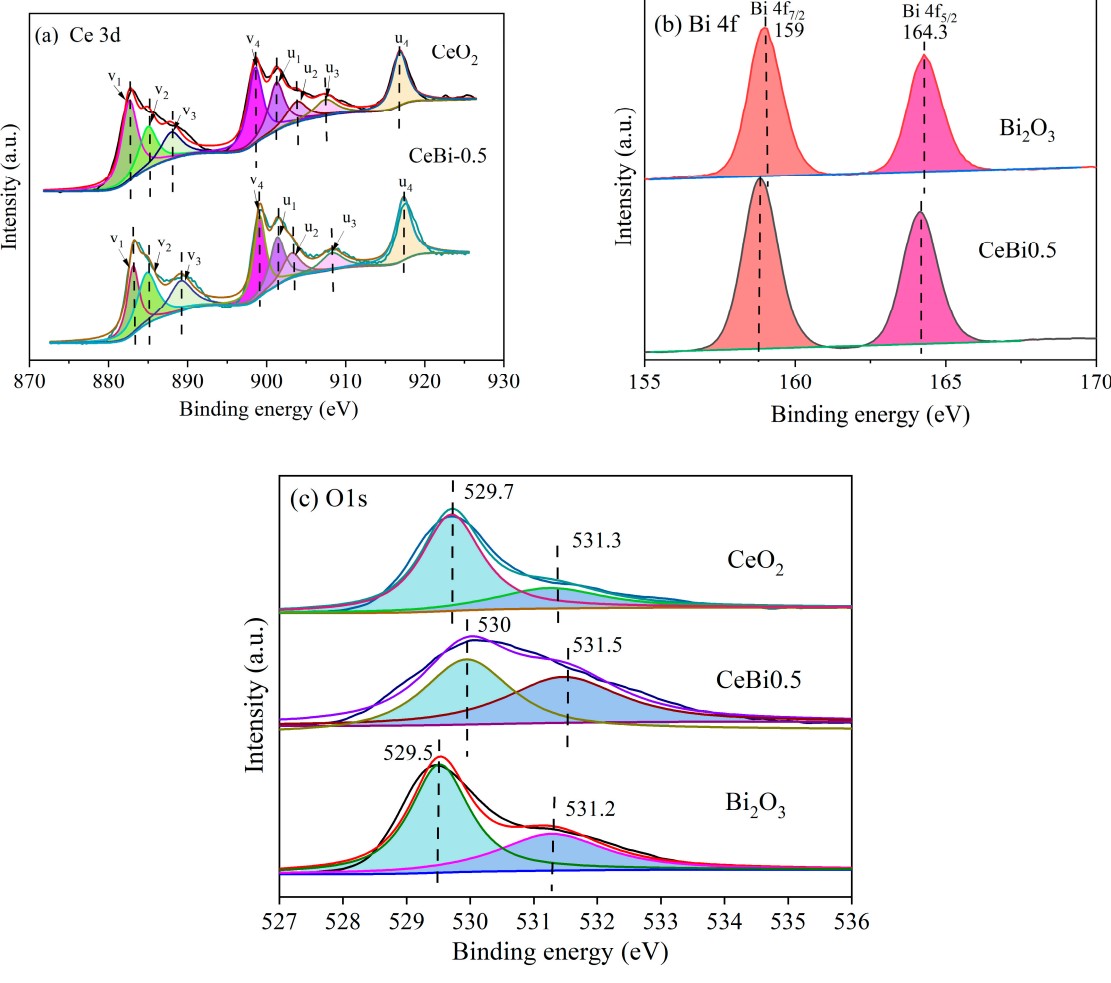

**Figure 5.** High-resolution XPS spectrum of (**a**) Ce 3d, (**b**) Bi 4f, (**c**) O 1s.

## 2.6. DRS Analysis

The UV-Vis DRS of $CeO_2$, $Bi_2O_3$ and $CeO_2/Bi_2O_3$ composites are shown in Figure 6a. Compared with the pure $CeO_2$ and $Bi_2O_3$, it is observed that the absorption edge of

CeO$_2$/Bi$_2$O$_3$ composites shifted significantly to the visible light region. In addition, the composites have higher absorption than pure CeO$_2$ in the visible light region, indicating that the heterojunction formed between CeO$_2$ and Bi$_2$O$_3$ could improve the light absorption ability and expand the light response region. The band gap energy of a semiconductor can be calculated by the following formula [42,43]:

$$(\alpha\,h\,v) = A(h\,v - E_g)^{n/2} \tag{1}$$

where $\alpha$, $h$, $n$, $E_g$ and A are absorption coefficient, Planck constant, light frequency, band gap energy and a constant, respectively, and n is 1 for a direct transition. Plotting $(\alpha h\,v)^2$ versus energy ($hv$) based on the spectral response in Figure 6a gives the extrapolated intercept corresponding to the $E_g$ value (Figure 6b). The band gap values ($E_g$) of pure CeO$_2$ and Bi$_2$O$_3$ are about 2.95 eV and 2.85 eV, respectively. Simultaneously, the $E_g$ of the CeO$_2$/Bi$_2$O$_3$ composites decreases gradually with the increase in the Bi$_2$O$_3$ molar ratio. Therefore, the prepared composites could make better use of visible light.

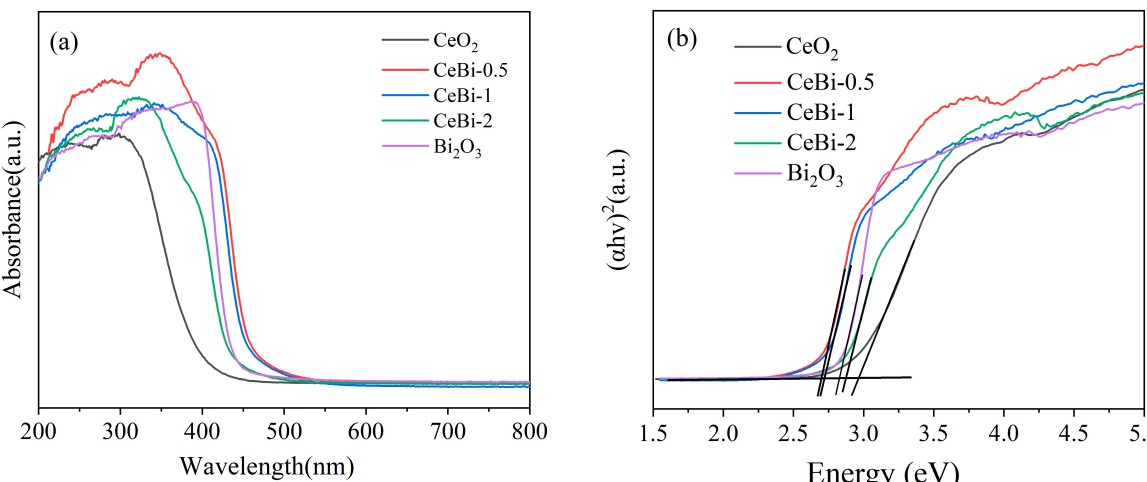

**Figure 6.** (**a**) UV-Vis DRS of CeO$_2$, Bi$_2$O$_3$ and CeO$_2$/Bi$_2$O$_3$ composites; (**b**) Tauc plots of $(\alpha hv)^2$ vs photon energy (hv).

### 2.7. Photocatalytic Activity

The removal experiment for DBT in model oil is carried out to study the photocatalytic activity of different photocatalysts under visible light. Figure 7a shows the photocatalytic desulfurization rate of CeO$_2$, Bi$_2$O$_3$ and CeO$_2$/Bi$_2$O$_3$ composites. It was found that all photocatalysts have a much lower adsorption ability for DBT in the dark. In the absence of any photocatalyst, the desulfurization rate of H$_2$O$_2$ is about 7.5% in 3 h, indicating that H$_2$O$_2$ can oxidize the DBT molecular to some extent, but the oxidation ability of H$_2$O$_2$ is extremely limited. When there is a photocatalyst in the reaction system, it can be found that the desulfurization rate is effectively improved. It is worth noting that the Bi$_2$O$_3$/CeO$_2$ composites show a significant desulfurization trend compared with the pure CeO$_2$ and Bi$_2$O$_3$. When the molar ratio of Ce to Bi is 1:2, the composite shows the best photocatalytic efficiency, and its desulfurization rate reaches 90.5 % in 3 h. Meanwhile, the corresponding photocatalytic desulfurization kinetic curves over the prepared photocatalysts are shown in Figure 7b. The reaction data are fitted by a first-order model as depicted by the formula [44]:

$$\ln(C_0/C_t) = k t \tag{2}$$

where $k$ is the pseudo-first-order rate constant, and the relationship between $\ln(C_0/C)$ and catalytic reaction time t is considered as linear. It is worth noting that CeBi-0.5 shows the maximum kinetic constant (k) value of 0.81802 h$^{-1}$, which is approximately 5.19-flod and 2.45-flod of pure CeO$_2$ and Bi$_2$O$_3$, respectively. This indicates that CeBi-0.5 composite can

effectively improve the photocatalytic activity. As shown in Figure 7c, the desulfurization rate of the CeBi-0.5 composite after three photodegradation cycles is 70%, which may be due to the loss of active components and the easy filling of the oxygen vacancy in the photocatalytic process [45]. Consequently, the excellent photocatalytic performance of the composite is attributed to its unique two-dimensional lamellar structure, which is conducive to the enrichment of organic sulfur molecules. Moreover, the formation of heterojunction between $CeO_2$ and $Bi_2O_3$ in the composite can promote the effective separation of photogenerated electrons and holes, and can significantly improve the photocatalytic activity.

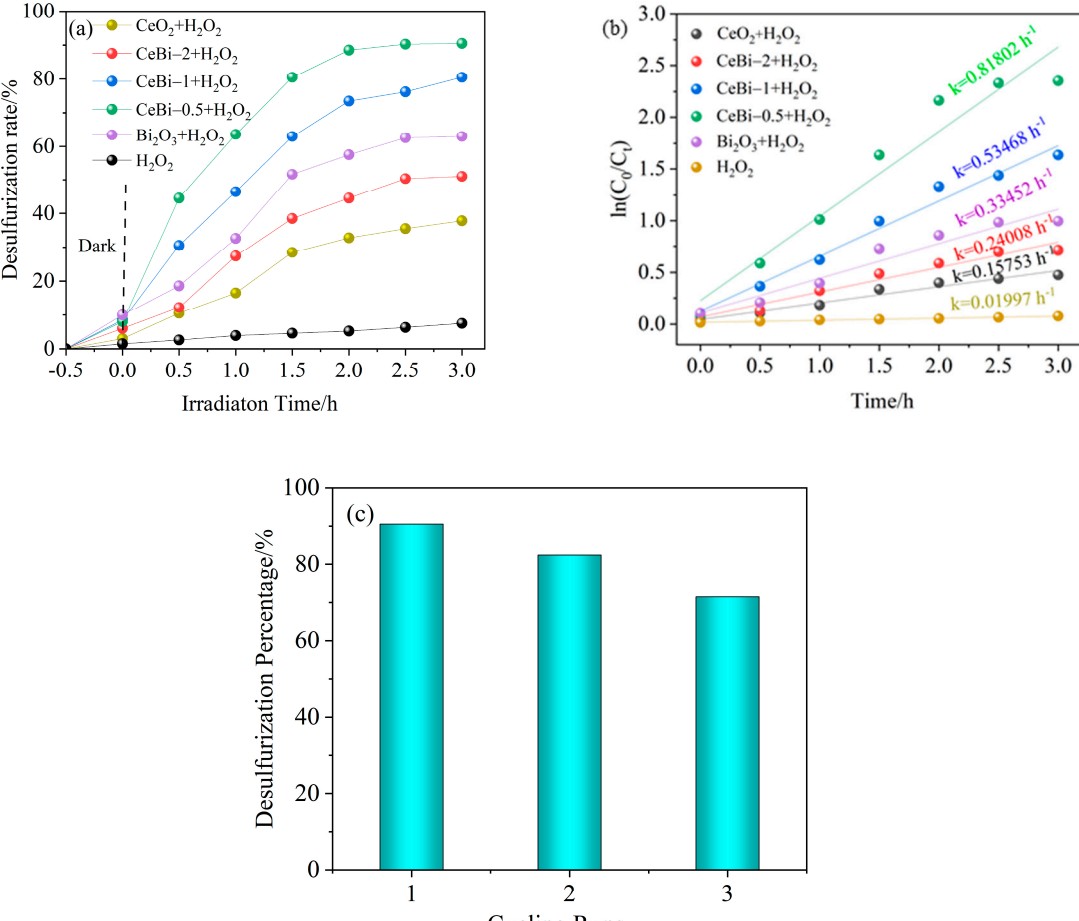

**Figure 7.** (**a**) Photocatalytic desulfurization rate of different catalysts; (**b**) Kinetic fitting curve; (**c**) Photocatalytic desulfurization cycle over the CeBi-0.5 composite.

### 2.8. Photoelectrochemical and PL Analysis

In general, the transient photocurrent responses could evaluate the separation and migration efficiency of photogenerated electron-hole pairs in the semiconductor materials The transient photocurrent response curve is shown in Figure 8a. The photocurrent intensity of CeBi-0.5 composite is larger than those of pure $CeO_2$ and $Bi_2O_3$, indicating that CeBi-0.5 composite exhibits the higher separation efficiency for electron-hole pairs [46]. Moreover, EIS is used to study the charge transfer resistance of the photogenerated carriers (Figure 8b). Generally speaking, the smaller the arc in the EIS Nyquist curve, the lower the charge transfer resistance on the electrode surface, and the recombination of photo generated charge carriers would be inhibited. The arc radius of CeBi-0.5 composite is significantly smaller than those of pure $CeO_2$ and $Bi_2O_3$, indicating that the CeBi-0.5 composite is beneficial for improving the transfer efficiency of photogenerated electron-hole pairs. Fluorescence spectroscopy is also an effective approach to study the separation efficiency of photogenerated electrons and holes. From Figure 8c, it can be found that the PL intensity

of CeBi-0.5 composite is lower than that of pure $CeO_2$ and $Bi_2O_3$, indicating that the recombination of photogenerated electrons and holes can be greatly inhibited in CeBi-0.5 composite [42]. Therefore, the formation of heterojunction structures between $CeO_2$ and $Bi_2O_3$ can effectively promote photo generation and electron separation, thereby improving photocatalytic activity.

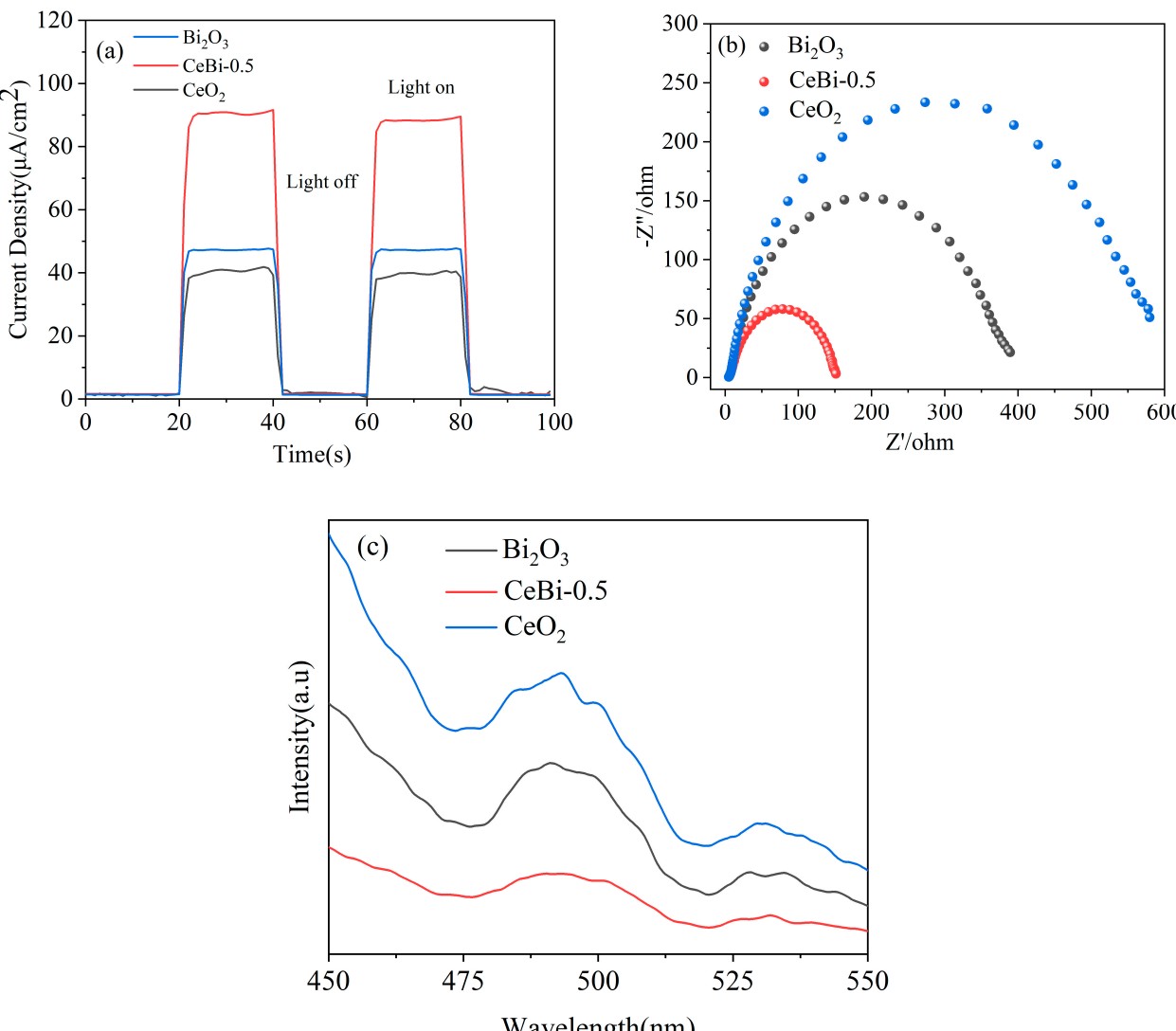

**Figure 8.** (**a**) Transient photocurrent responses curve, (**b**) Nyquist impedance plots, (**c**) PL spectra of $CeO_2$, $Bi_2O_3$ and CeBi-0.5 composite.

### 2.9. Photocatalysis Mechanism

In general, photoinduced holes ($h^+$), superoxide radicals ($\cdot O^{2-}$) and hydroxyl radicals ($\cdot OH$) are considered to be the main active species in the photocatalytic degradation of semiconductors [19]. In order to further determine the main active species of $CeO_2/Bi_2O_3$ composites in photocatalytic reactions, the trapping capture experiment for active species is carried out. Three different kinds of captors, triethanolamine (TEOA; $h^+$ scavenger), tert-butyl alcohol (TBA; a $\cdot OH$ radical scavenger) and benzoquinone (BQ; a $\cdot O^{2-}$ radical scavenger), are used [47]. As can been seen in Figure 9, the desulfurization rate is slightly decreased using TEOA and BQ as capture agents, indicating that neither $h^+$ nor $\cdot O^{2-}$ is the main reactive species. However, the addition of TBA causes obvious deactivation of the photocatalyst, reducing the photocatalytic activity for the desulfurization rate from 90.5% to 35%, as shown in the experimental results. This clearly demonstrates that active $\cdot OH$

radicals are the dominant reactive species responsible for the photocatalytic desulfurization system over these photocatalysts.

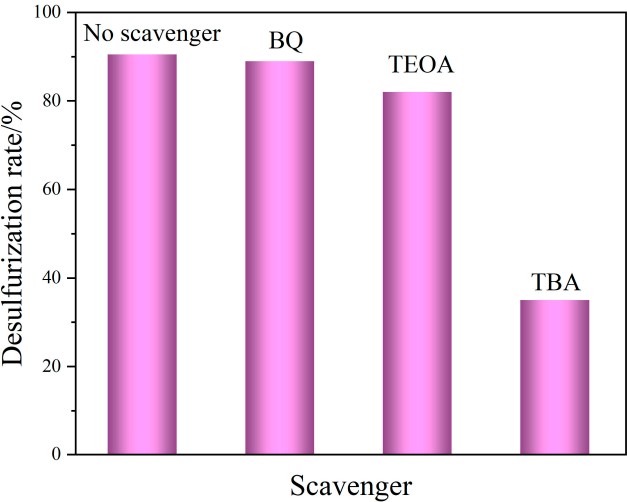

**Figure 9.** Photocatalytic desulfurization by different scavengers over CeBi-0.5.

To further determine the mechanism of photocatalytic desulfurization, the positions of conductive band (CB) and valence band (VB) need to be determined, according to the following formula [48,49]:

$$E_{CB} = \chi - E_0 - 0.5E_g \tag{3}$$

$$E_{VB} = E_{CB} + E_g \tag{4}$$

where $\chi$ is the absolute electronegativity of the semiconductor (the $\chi$ values of $Bi_2O_3$ and $CeO_2$ are 5.99 eV and 5.56 eV, respectively.), $E_0$ is the potential energy of the standard hydrogen electrode (4.5 eV), and $E_g$ is the band gap of the semiconductor. Combined with $E_g$ values of $Bi_2O_3$ and $CeO_2$ are 2.85 eV and 2.95 eV, and the CB and VB values of $Bi_2O_3$ are calculated to be 0.07 eV and 2.92 eV, and −0.42 eV and 2.53 eV for $Bi_2O_3$.

The mechanism of photocatalytic desulfurization is shown in Figure 10. When the composite is irradiated under visible light, both $Bi_2O_3$ and $CeO_2$ can be activated and generate electron-hole pairs, due to the CB of $CeO_2$ being more negative than that of $Bi_2O_3$. The photogenerated electrons on the CB of $CeO_2$ tend to migrate to that of $Bi_2O_3$ through the interface, and then the electrons will react with $H_2O_2$ to form ·OH, whereas the holes in the VB of $Bi_2O_3$ are spontaneously transferred to $CeO_2$, where -OH can be oxidized to ·OH by the holes. Hydroxyl radicals have strong oxidation ability, which can oxidize non-polar DBT molecules adsorbed on the surface of two-dimensional materials into polar dibenzothiophene sulfone ($DBTO_2$). Then $DBTO_2$ can be extracted and removed, due to its strong polarity, and finally the organic sulfur in the model oil is effectively removed [50]. Based on the above discussion, we propose the following reaction equations:

$$CeO_2/Bi_2O_3 + h\nu \rightarrow e^- + h^+ \tag{5}$$

$$e^- + H_2O_2 \rightarrow \bullet OH + OH^- \tag{6}$$

$$h^+ + OH^- \rightarrow \bullet OH \tag{7}$$

$$DBT + \bullet OH \rightarrow DBTO_2 \tag{8}$$

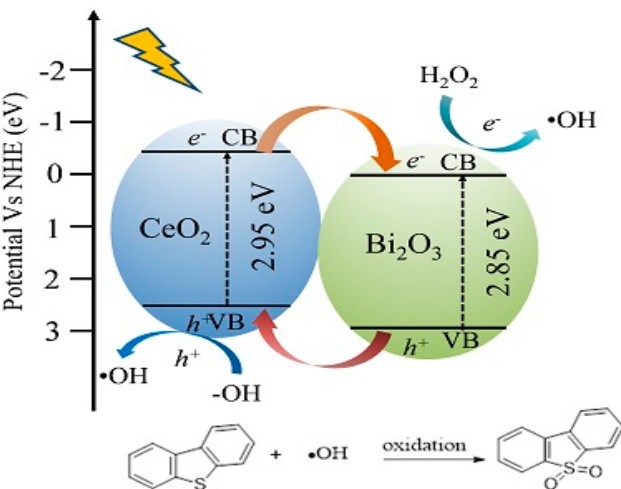

**Figure 10.** Photocatalytic desulfurization mechanism of $CeO_2/Bi_2O_3$ nanocomposite.

## 3. Experimental Section

### 3.1. Materials

Cerium nitrate hexahydrate ($Ce(NO_3)_3 \cdot 6H_2O$, 99%),Bismuth nitrate pentahydrate ($Bi(NO_3)_3 \cdot 5H_2O$, 99%), Acetonitrile ($C_2H_3N$, 99%), Anhydrous ethanol ($CH_3CH_2OH$,) and Tert-butyl alcohol (($CH_3)_3COH$, TBA) were purchased from China National Medicines Corporation Ltd. Benzo-thiophene ($C_{12}H_8S$, DBT) and n-octane ($C_8H_{18}$) were purchased from Aladdin Bio-Chem Technology Co., Ltd. (Shanghai, China). Ethylene glycol (($CH_2OH)_2$), Triethanolamine ($HOCH_2CH_2)_3N$, TEOA) and Benzoquinone ($C_6H_4O_2$, BQ) were purchased from Shanghai Maclean Biochemical Technology Co. Ltd. All of the reagents in this experiment were analytical grade and directly used without further purification. Distilled water was used throughout this study.

### 3.2. Preparation of Samples

The typical synthesis process is described as follows: 0.01 mol $Bi(NO_3)_3 \cdot 5H_2O$ was dissolved in a mixed solution of 60 mL ethylene glycol and ethanol ($V_{EG}$: $V_{Ethanol}$ = 1:2) marked as A, and then a certain stoichiometric ratio of $Ce(NO_3)_3 \cdot 6H_2O$ was dissolved in 10 mL deionized water, marked as B. Subsequently, solution B was slowly dropped into solution A, After the two solutions were fully mixed, the solution was relocated into a 100 mL Teflon liner and then placed in a microwave synthesizer at 160 °C for 30 min, then the precipitate was washed with distilled water and anhydrous ethanol three times, and dried in an oven at 60 °C for 12 h. The composites with molar ratios between $CeO_2$ and $Bi_2O_3$ (2:1, 1:1, 1:2) were labeled as CeBi-2, CeBi-1 and CeBi-0.5, respectively. Pure $CeO_2$ and $Bi_2O_3$ photocatalysts were also synthesized under similar experimental conditions.

### 3.3. Characterization

X-ray diffraction (XRD) measurement was performed by a Rigaku, D/max-RB instrument (Tokyo, Japan) between 20 and 80° at a scan rate of 5°/min with Cu Ka radiation; Raman spectra were measured by a Thermo Fisher Scientific DXR spectrometer (Waltham, MA, USA) and the excitation laser wavelength was 532 nm. The morphology was observed by field emission scanning electron microscopy (FESEM, Zeiss Supra55, Jena, Germany) and transition electron microscopy (TEM, JEOL JEM-2100, Tokyo, Japan), equipped with an EDS spectrometer (EDS, Oxford, UK). $N_2$ adsorption-desorption isotherms were determined using a Micromeritics ASAP 2010 analyzer (Norcross, GA, USA), the surface areas were calculated according to the Brunauer-Emmett-Teller (BET) method, and the pore size distribution was obtained by the Barrett-Joyner-Halenda (BJH) model. X-ray photoelectron spectroscopy (XPS) analysis was carried out by a Thermo Fisher Scientific ESCALAB 250 (Waltham, MA, USA) spectrometer with mono Al Ka radiation (1486.6 eV). Photoluminescence (PL) spectra were collected on a PerkinElmer LS45 fluorescence spectrometer

(Waltham, MA, USA). Ultraviolet-visible diffuse reflectance spectra (UV-Vis DRS) were measured by a Shimadzu UV-2450 spectrophotometer (Kyoto, Japan) equipped with an integrating sphere.

### 3.4. Photocatalytic Desulfurization Measurement

The model oil with sulfur content of 100 ppm was prepared by dissolving 0.08 g DBT into 200 mL of n-octane. Then, 0.1 g of photocatalysts were added to the above solution with constant stirring; subsequently, an appropriate 30wt% hydrogen peroxide aqueous solution (molar ratio of ($H_2O_2$/DBT = 4:1) was added. After dark adsorption for 30 min, the suspension was transferred into a photochemical reaction instrument containing a 300 W xenon lamp with an ultraviolet cut-off filter (GHX-2, Yangzhou Science and Technology City Instrument co., Ltd., China). The dispersion was collected every 30 min and extracted with acetonitrile, then the sulfur content was measured using an ultraviolet fluorescence sulfur analyzer (THA2000S, Taizhou Jinhang Analysis Instrument Co., Ltd., Taizhou, China). The desulfurization rate D (%) is obtained according to the following formula:

$$D = (1 - C_t/C_0) \times 100\% \tag{9}$$

where $C_0$ is the initial sulfur content and $C_t$ is the sulfur content of the solution at reaction time t.

### 3.5. Photoelectrochemical Measurements

Photoelectrochemical analysis was performed in a standard three electrode system, with Pt foil as the counter electrode, Ag/AgCl electrode as the reference electrode and the prepared sample deposited on FTO glass as the working electrode. Photocurrent was measured on a photoelectrochemical workstation (LK5600, Tianjin Lanlike Chemical Electronics High Technology Co., Ltd., Tianjin, China) with 0.1 M $Na_2SO_4$ aqueous solution as electrolyte and 300 W xenon lamp as light source. EIS test also used the three-electrode system. The electrolyte solution was 0.5 M KCl containing 0.01 M $K_3Fe(CN)_6$/$K_4Fe(CN)_6$ (molar ratio 1:1), performed at bias voltages 0.5 V, in the frequency range of 0.1 Hz-100 kHz, with oscillation potential amplitudes of 0.01 V.

### 4. Conclusions

In summary, a two-dimensional layered $CeO_2$/$Bi_2O_3$ photocatalyst was successfully synthesized by microwave solvothermal method and used for photocatalytic desulfurization. The prepared photocatalysis significantly enhances the performance of photocatalytic removal of DBT under visible light irradiation. When the molar ratio of $Bi_2O_3$ to $CeO_2$ is 2:1, the prepared sample exhibits the best photocatalytic activity, with removal rate of DBT reaching 90.5% in3 h and the reaction rate constant at 0.81802 $h^{-1}$, which is approximately 5.19-flod and 2.45-flod of pure $CeO_2$ and $Bi_2O_3$, respectively. Free radical scavenging experiments demonstrate that hydroxyl radical is the main active species in the photocatalytic process. The enhanced photocatalytic activity is attributed to the unique two-dimensional structure of the heterojunction formed between cerium oxide and bismuth oxide. This study indicates that the construction of two-dimensional heterostructure photocatalyst is a promising method to improve photocatalytic desulfurization.

**Author Contributions:** Conceptualization, X.L. and Q.Z.; methodology, X.L. and J.Q.; validation, X.L. and H.H.; formal analysis, X.L., W.C. and H.H.; investigation, X.L. and W.C.; resources, X.L.; data curation, H.H. and W.C.; writing-original draft preparation, X.L. and W.C.; writing-review and editing, H.H. and J.Q.; visualization, X.L. and H.H.; supervision, X.L.; project administration, Q.Z.; funding acquisition, X.L. and Q.Z. All authors have read and agreed to the published version of the manuscript.

**Funding:** This work is supported by the National Natural Science Foundation of China (No. 12274361), the Natural Science Foundation of Jiangsu Province (BK20211361), College Natural Science Research Project of Jiangsu Province (20KJA430004) and the Funding for school-level research projects of Yancheng Institute of Technology (xjr2019026).

**Data Availability Statement:** Data available on request from the authors.

**Conflicts of Interest:** The authors declare no conflict of interest.

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
