# Peer review of "Preparation of Two-Dimensional Layered CeO2/Bi2O3 Composites for Efficient Photocatalytic Desulfurization"

_catalysts, doi:10.3390/catal13050821_

Round 1

Reviewer 1 Report

The article is of significant scientific and practical interest in terms of creating new photocatalytic materials based on cerium and bismuth. However, due to the fuzzy description of the experimental procedure, the effectiveness of the developed photocatalysts is questionable. The article can be published after the following remarks are eliminated:

1. There are questions to the description of the experimental methodology that mislead the reader:

1.1.Explain in more detail what "initial sulfur content" is. Is this the concentration of dibenzothiophene? Is this the concentration of sulfur in dibenzothiophene at its known concentration?

1.2. Correct and complete the sentence (line 513-514). It is not clear what the authors mean by indicating the molar ratio:

- specify units for 0.1

-The developed photocatalysts also contain oxygen, and then the molar ratio is (photocatalyst + hydrogen peroxide)/(dibenzothiophene) or (hydrogen peroxide)/(dibenzothiophene)?

 Then  0.1  photocatalysts  and  appropriate  hydrogen peroxide aqueous   (molar ratio of O/S = 8:1) were added.

1.3. A photocatalytic process is 3-4 parallel processes:

item 1. Sorption of the pollutant on the photocatalyst (the authors took into account this process - line: 514)

item 2. Self-decomposition of a pollutant under the action of light without a photocatalyst (the authors do not show this process in Fig. 7 a, b)

item 3. Decomposition of a pollutant under the action of a photocatalyst (the authors show this process in Figure 7 a, b)

item 4. Decomposition of the pollutant under the action of other substances in the photocatalytic solution (the authors use hydrogen peroxide at a concentration 8 times higher than the concentration of sulfur / dibenzothiophene, however, in Figure 7 a, b, the oxidation of dibenzothiophene under the action of hydrogen peroxide is not indicated)

For example, the authors from the reference used in the work [17] in Figure 10 show the decomposition of the substrate in the presence of hydrogen peroxide

Conclusion: lack of data in figure 7 a, b according to item 2. and item 4. does not allow an objective comparison of the efficiency of the developed photocatalysts.

1.4. How do the authors explain the choice of DIbenzothiophene as a substrate for photocatalytic studies? For example: In their previous work (17) the authors used benzothiophene.

Dibenzothiophene is a model substance in studies of photocatalytic desulfurization?

Is dibenzothiophene the most common sulfur contaminant in various oils?

Explain in more detail in the introduction or in the experimental procedure the choice of dibenzothiophene.

2. Benzothiophene and dibenzothiophene are two different chemical compounds!

Specify clearly the connection and its abbreviation used in the work:

  - by removing dibenzothiophene (DBT) line 86

- oxidation of benzothiophene (BT) line 63

- benzothiophene (C 12 H 8 S, DBT) and n-octane line 479

3. The use of abbreviations is intended to shorten the text of the article. Why use abbreviations multiple times?

For example:

spectroscopy (EIS) test ...line 530

electrochemical impedance spectroscopy (EIS) ...line 19

Electrochemical impedance spectroscopy (EIS) ...line 384

When many abbreviations are used, it is most correct to create a single list of abbreviations.

4. Get your formulas in order:

- line 311 Numbering available

- line 340 is not numbered. formula inside text

- line 440-441 No numbering

- line 521 is not numbered. formula inside text

Minor editing of English language required

Author Response

Reviewer 1

1.There are questions to the description of the experimental methodology that mislead the reader:

1.1. Explain in more detail what "initial sulfur content" is. Is this the concentration of dibenzothiophene? Is this the concentration of sulfur in dibenzothiophene at its known concentration?

Responses 1: Thanks for the reviewer’s suggestion. We apologize for any confusion caused and appreciate the valuable suggestions. " Initial sulfur content", Which is the concentration of sulfur in n-octane. The molecular weight of sulfur and dibenzothiophene is 32 and 184.257. respectively. The density of n-octane is 0.7 g/ml, Therefore, to prepare a 200ml n-octane solution with a sulfur content of 100 ppm, the required mass of dibenzothiophene can be calculated:

1.2. Correct and complete the sentence (line 513-514). It is not clear what the authors mean by indicating the molar ratio:

- specify units for 0.1

-The developed photocatalysts also contain oxygen, and then the molar ratio is (photocatalyst + hydrogen peroxide)/(dibenzothiophene) or (hydrogen peroxide)/(dibenzothiophene)?

 Then 0.1 photocatalysts and appropriate hydrogen peroxide aqueous (molar ratio of O/S = 8:1) were added.

Responses 2: Thank you for your useful comment. We apologize for any confusion caused and appreciate the valuable suggestions.We describe the experimental process in detail.”The model oil with sulfur content of 100 ppm was prepared by dissolving 0. 08 g DBT into 200 mL of n-octane.Then 0.1g photocatalysts were added to the above solution at constant stirring, subsequently, an appropriate 30wt% hydrogen peroxide aqueous solution (molar ratio of (hydrogen peroxide)/(dibenzothiophene = 4:1) were added.”, the molar ratio hydrogen peroxide and dibenzothiophene is 4 : 1, so the molar ratio of O to S is 8 : 1.

1.3. A photocatalytic process is 3-4 parallel processes:

item 1. Sorption of the pollutant on the photocatalyst (the authors took into account this process - line: 514)

item 2. Self-decomposition of a pollutant under the action of light without a photocatalyst (the authors do not show this process in Fig. 7 a, b)

item 3. Decomposition of a pollutant under the action of a photocatalyst (the authors show this process in Figure 7 a, b)

item 4. Decomposition of the pollutant under the action of other substances in the photocatalytic solution (the authors use hydrogen peroxide at a concentration 8 times higher than the concentration of sulfur / dibenzothiophene, however, in Figure 7 a, b, the oxidation of dibenzothiophene under the action of hydrogen peroxide is not indicated)

For example, the authors from the reference used in the work [17] in Figure 10 show the decomposition of the substrate in the presence of hydrogen peroxide

Conclusion: lack of data in figure 7 a, b according to item 2. and item 4. does not allow an objective comparison of the efficiency of the developed photocatalysts.

Responses 3:

For item 1. Thanks for the reviewer’s suggestion. The time is -30 minutes to 0 minutes, without light radiation, indicating the process of dark adsorption. We have marked in Figure 7 ( a ) now. and the description of dark adsorption is added in line 317-322.

For item 2. Thank you for your rigorous consideration. As a comparison, we added photocatalytic desulfurization experiments with only hydrogen peroxide and no photocatalyst. the experimental results are shown in Figure 7 ( a ), ( b ), now.

For item 3. Thank you for the reviewer’s comment. We use the ultraviolet fluorescence sulfur analyzer to measure the sulfur content, then calculate the degradation, and finally draw the degradation rate curve.

For item 4. Thanks for the reviewer’s suggestion. We added photocatalytic desulfurization experiments with only hydrogen peroxide and no photocatalyst. the experimental results are shown in Figure 7 ( a ), ( b ), now.

   At present, hydrogen peroxide is mainly used as photocatalytic desulfurization oxidant in the literature.We have considered the use of oxygen, ozone as an oxidant, However, the gas is introduced into the reactor, and the requirements for the equipment are high ( closed containers are required, and the flow rate and flow rate need to be controlled ).our experimental conditions can not be achieved. In the future, we will try some new oxidation aids

   Thank you for the reviewer’s comment. In Conclusion, We have added data based on Figure 7 ( a ) ( b )

1.4. How do the authors explain the choice of DIbenzothiophene as a substrate for photocatalytic studies? For example: In their previous work (17) the authors used benzothiophene.

Dibenzothiophene is a model substance in studies of photocatalytic desulfurization?

Is dibenzothiophene the most common sulfur contaminant in various oils?

Explain in more detail in the introduction or in the experimental procedure the choice of dibenzothiophene.

Responses 4: Thanks for the reviewer’s suggestion.

Dibenzothiophene (DBT) and its derivatives are the major sulfur species in diesel and gasoil. Difficulties in treating the DBT molecules arise from the steric hindrances and stabilization of the S-C bond within the molecule. Deep desulfurization of these thiophene molecules can be achieved by HDS at high reaction temperatures and pressures, with 5- to 15-fold larger process units required ( J. Environ. Chem Eng, 2014,2(4), 1947-1955; Energy Procedia, 2015,74, 663-678; Appl. Catal., B, 2022,316, 121614.). Moreover, dibenzothiophene (DBT) which accounts 70% of the sulfur compounds in diesel using a synthetic and typical South African diesel ( Sci. Rep., 2023,13(1), 6020. ). Therefore, the removal of dibenzothiophene is more difficult than that of benzothiophene.

It is more meaningful to choose dibenzothiophene as a model substance in studies of photocatalytic desulfurization Dibenzothiophene (DBT) and its derivatives are the major sulfur species in diesel and gas oil.

In the introduction, line 34-38, we have added an introduction to dibenzothiophene.

2.. Benzothiophene and dibenzothiophene are two different chemical compounds!

Specify clearly the connection and its abbreviation used in the work:

-by removing dibenzothiophene (DBT) line 86

-oxidation of benzothiophene (BT) line 63

- benzothiophene (C12H8S, DBT) and n-octane line 479

Responses5: Thanks for your careful checks. We are sorry for our carelessness. Line 479 benzothiophene (C12H8S, DBT) is a mistake, and change to dibenzothiophene (C12H8S, DBT )

  1. The use of abbreviations is intended to shorten the text of the article. Why use abbreviations multiple times?

For example:

spectroscopy (EIS) test ...line 530

electrochemical impedance spectroscopy (EIS) ...line 19

Electrochemical impedance spectroscopy (EIS) ...line 384

When many abbreviations are used, it is most correct to create a single list of abbreviations.

Responses 6: We gratefully appreciate for your valuable suggestion. We revised and used the full name when it first appeared in the article, followed by abbreviations.

  1. Get your formulas in order:

- line 311 Numbering available

- line 340 is not numbered. formula inside text

- line 440-441 No numbering

- line 521 is not numbered. formula inside text

Responses 7: Thank you for your rigorous comment. We have numbered the formulas in order.

Reviewer 2 Report

Review Comments: catalysts-2354480

In this manuscript, authors have prepared a two-dimensional layered CeO2/Bi2O3 composite using microwave solvothermal method. The activity of the photocatalysts were evaluated by removing dibenzothiophene (DBT) from the model oil. In addition, the crystal structure, morphology, optical performance , elemental composition and surface electronic state of the prepared samples were investigated by various characterization techniques. Although, study is interesting, however, in the present form it has several issues which should be properly addressed. Following are the comments to be addressed:

1-Figure 1(XRD): Why some peaks have been labeled and others are not?

2-Introduction section: Authors have described several hydrogen-free desulfurization technologies such as oxidative desulfurization (ODS), adsorptive desulfurization (ADS), extractive desulfurization (EDS)  and biological desulfurization (BDS). I think still here is some gap of description of the latest alternative technologies of HDS? For instance, authors should also mention about Photocatalytic Oxidative Desulfurization, Deep Aerobic Photocatalytic Oxidtive Desulfurization and etc. There are several latest literature is available regarding this in some famous journals like Catalysis Reviews, Chemical Engineering Journal, Journal of Industrial and Engineering Chemistry, Nanomaterials, Journal of Cleaner Production and etc. Authors should strengthen this part with relevant references from these journals.

3-Figure 9: It shows the use of different scavengers have even reduced the activity, why? In literature there are several scavenging agents have been reported which are very supportive for ODS, for example “Chem. Eng. J. 2022,441, 136063”. Please check and improve accordingly.

4-Regarding the photocatalytic activity, authors should check the activity under dark conditions to see any effect of adsorption. Also, under light without using any catalyst?

5-All equations used in this manuscript should be properly numbered and cited in the text.

Overall English is okay, only minor grammatical errors are there in a few places. Authors may carefully review the manuscript. 

Round 2

Reviewer 1 Report

Notes have been corrected. The article can be published in its current form

Author Response

Thank you.

Reviewer 2 Report

Authors have properly addressed all the comments and the revised manuscript is now recommended to be accepted.

Author Response

Thank you.